# Smellscape Characteristics of an Urban Park in Summer: A Case Study in Beijing, China

**Chen Wang, Ruolin Zhu, Jian Zhong** **, Huajin Shi, Chang Liu, Huiyu Liu, Bohao Tan, Lijuan Xiang, Ruizi Xiang, Xinru Ye and Ming Sun ***

State Key Laboratory of Efficient Production of Forest Resources, Beijing Key Laboratory of Ornamental Plants Germplasm Innovation and Molecular Breeding, National Engineering Research Center for Floriculture, Beijing Laboratory of Urban and Rural Ecological Environment, Key Laboratory of Genetics and Breeding in Forest Trees and Ornamental Plants of Ministry of Education, School of Landscape Architecture, Beijing Forestry University, Beijing 100083, China; wangchenbjfu@163.com (C.W.); ruolinzhu@163.com (R.Z.); zhongjianbjfu@163.com (J.Z.); shihuajinbjfu@163.com (H.S.); liuchangbjfu@163.com (C.L.); liuhuiyubjfu@163.com (H.L.); tanbohaobjfu@163.com (B.T.); xianglijuanbjfu@163.com (L.X.); xiangruizibjfu@163.com (R.X.); yexinrubjfu@163.com (X.Y.)
* **Correspondence: sunmingbjfu@163.com**

**Abstract:** The construction of urban green spaces is a pivotal aspect of sustainable urban development. As societal preferences evolve, a shift from visually oriented landscapes to multi-sensory landscapes has emerged. However, scant attention has been given to the olfactory dimension of urban green spaces. This study addresses this gap by investigating the relationship between odor perception and park visit experiences, employing a combination of smellwalks and questionnaire surveys conducted in Purple Bamboo Park in Beijing. Natural odors, with most perception frequencies above 60%, are the most dominant odors in Purple Bamboo Park during the summer, including plant, water, and soil smells. The questionnaire survey results revealed a positive correlation between the perception of natural odors and tour experience. Notably, floral fragrances emerged as the predominant olfactory stimulus influencing the park's olfactory ambiance. Furthermore, a remarkably strong association was observed between the degrees of olfactory, visual, and overall experiential satisfaction, which indicates that multi-sensory experiences in urban parks work as an organic whole. By recognizing the pivotal role of smell in shaping perceptions, urban planners and designers can now integrate olfactory considerations into their work, thereby elevating the overall quality and sustainability of urban green spaces.

**Keywords:** smellscape; smellmap; multi-sensory; visiting satisfaction; urban park

## 1. Introduction

Urban green spaces are crucial for sustainable urban development [1]. As urbanization accelerates globally, the significance of these spaces in mitigating environmental degradation and enhancing quality of life becomes increasingly evident. On the one hand, urban green spaces can purify the air, alleviate the heat island effect, and play an essential role in ecological functions [2–5]. On the other hand, they provide leisure and recreational places for urban residents and are indispensable in constructing urban habitats [6].

The senses play an important role in shaping the relationship between people and the environment and form an integrated knowledge-gathering system [7,8]. Different types of sensory information can enrich people's interactions with places and their social interactions. Experiences with abundant sensory information enhance people's memories of a place and help form the identity of an area [9]. With the development of society, people are no longer satisfied with vision-centered landscaping. The concept of multi-sensory gardens thus emerged, which combines vision, touch, smell, sound, and taste [10–13]. Research on visual landscapes is now relatively well established, including the study of visual quality

assessments and visual landscapes for psychological well-being and ecology [14–16]. There are also many studies of soundscapes, such as those on the classification and evaluation of soundscape, the effects of soundscape on public visiting experience, and soundscape improvement and design [17–21]. Compared with research on visual landscapes and soundscapes, research on smell, taste, and touch is rare.

The "Taste–Smell System" is one of the five basic systems of human perception of an environment [22]. The human olfactory system, with its hundreds of different olfactory receptors, far outperforms the other senses in the number of physically different stimuli it can discriminate [23]. People are inevitably exposed to air and odors as they walk through a city. Therefore, smell is a necessary part of the urban sensory experience. The concept of smellscape was first raised by Canadian geographer Porteous in 1985 with reference to the definition of soundscape as a cross-field that discusses the relationship among humans, odor, and the environment [24]. Porteous drew an analogy between olfaction and vision and recognized the value of olfaction as an aesthetic subject. A smellscape aims to arouse olfactory enthusiasm among the public as well as urges the recording, protection, recurrence, and creation of a more harmonious olfactory environment. Beyond the visual and auditory elements traditionally associated with design, the incorporation of olfactory sensations, as highlighted by the concept of a smellscape, adds a unique and influential dimension to the overall experience.

(1)   Enhancing Emotional Connection

Olfactory sensations are known to evoke powerful emotional responses [25]. The inclusion of smells in public spaces can trigger nostalgia, calmness, or excitement, creating a more profound and memorable connection between individuals and their surroundings. Moreover, shared olfactory experiences can facilitate social interactions. Aromatic elements in public spaces can serve as conversation starters and bring people together, fostering a sense of community and shared experience [9]. Odor perception in the environment both increases people's interaction with the environment and promotes human social interaction.

(2)   Cultural and Identity Significance

Scents often carry cultural and regional significance. Introducing culturally relevant smells to public spaces can help express local identity and heritage. For example, the aroma of traditional cuisine or indigenous plants can contribute to a sense of place and community [26]. In the context of cultural tourism, smellscape becomes a crucial element. Highlighting local scents in tourist destinations can provide a more immersive experience, allowing visitors to connect with the destination on a deeper level through their sense of smell. Case studies have been undertaken for classical gardens, affirming the cultural significance of odors and providing valuable insights for crafting olfactory landscapes in modern garden designs [27,28]. There is a campaign of 100 examples of selected smellscapes in Japan, covering a wide range of smellscapes such as plant scents, food scents, and incense scents, recognizing the value of scent as a landscape element [29]. Different odors play an essential role in urban culture, and the study of smellscapes can contribute to the preservation of urban culture and the establishment of residents' sense of belonging in the urban dimension.

(3)   Fostering Well-being and Comfort

Pleasant smells have the potential to enhance well-being and comfort. Incorporating natural and calming scents, such as those from gardens or green spaces, can contribute to a more relaxing and enjoyable atmosphere. The effects of aromatic compounds on the brain in multi-sensory environmental interventions are thought to be directly mediated. Thus, smellscape is considered an important healing element in horticultural therapy, applied to comprehensive health outcomes including mental health, physical health, reduced agitation behavior, improved cognitive function, and improved well-being [30,31].

(4)    Navigation and Spatial Awareness

Smells can also serve as unique markers in spatial navigation. Certain scents may signify specific areas or activities within a public space, aiding individuals in orientation and creating a more intuitive and navigable environment.

Thus, smellscape integration into the design of public spaces adds layers of meaning and richness to the overall sensory experience. Recognizing the significance of olfactory sensations alongside visual and auditory elements contributes to more holistic, engaging, and culturally resonant environment creation for diverse communities. Smellwalk is the main method of conducting smellscape background surveys and refers to the process by which a group of investigators sniff, capture, and record smells in the study area [32,33]. The investigation results are presented through perception models and smell mapping, which can provide a reference for urban design [34]. An influx of research cases has recently arisen, with the research scope continually broadening. Studies at the street scale have been executed for the old city of Beijing and the streets in Guangdong to probe into odor distribution and its influence on cultural tourism [35,36]. On the urban scale, a smellwalk conducted in Kastamonu City exposed the significance of smell in urban memory and finally emphasized the importance of smell for urban design and planning [37]. In the book "Urban Smellscape", a comprehensive elucidation of urban odor perception and various urban odor types is provided, along with an emphasis on the importance of odor perception in the overall perception of a city [38]. In addition to streets and urban areas, researchers have realized the lack of olfactory experiences in the design of architectural spaces and attempted to explore the impact of smells on architecture [39].

While extensive research has delved into urban smellscapes within streets and architectural settings, there remains a notable dearth in studies centered on urban park environments [40,41]. Additionally, the realm of olfactory research methodology reveals a notable gap in the availability of quantitative metrics for appraising odor perception. This gap hinders the precision and comparability of findings across studies. Hence, there arises a crucial need for the development and standardization of such metrics to bolster the rigor and depth of olfactory research in diverse urban contexts. To fill this gap, this study focused on the urban park environment and attempts to establish an evaluation system for smellscapes. Therefore, we expanded the survey population and chose Purple Bamboo Park in Beijing as a case to reveal the relationships between smell perception and public visiting experience in urban parks. This study will enhance our understanding of smellscapes and help us to improve the olfactory landscapes in urban parks.

## 2. Materials and Methods

### 2.1. Study Area

Beijing, the capital of the People's Republic of China, has a typical semi-humid continental monsoon climate with a hot, rainy summer. The research site, Purple Bamboo Park (116°19′ E, 39°56′ N), is located in the Haidian District of Beijing, covering an area of about 47.4 hectares (Figure 1). Purple Bamboo Park was built in 1953 with typical characteristics of Chinese classical gardens. It was named after a centuries-old temple, Purple Bamboo Temple in Bliss, and covers a variety of attractions, such as Poem for Lane of Mottled Bamboos, Bright Moon Islet, and Garden of Graceful Stones.

Bodies of water account for a third of the park. Two rivers pass through the park, forming a basic pattern of three lakes, two islands, a dike, a river, and a canal. Moreover, the park is dominated by naturalistic landscaping with large areas of bamboo and aquatic plants, which is a major feature of the park and provides a wonderful travel experience for visitors.

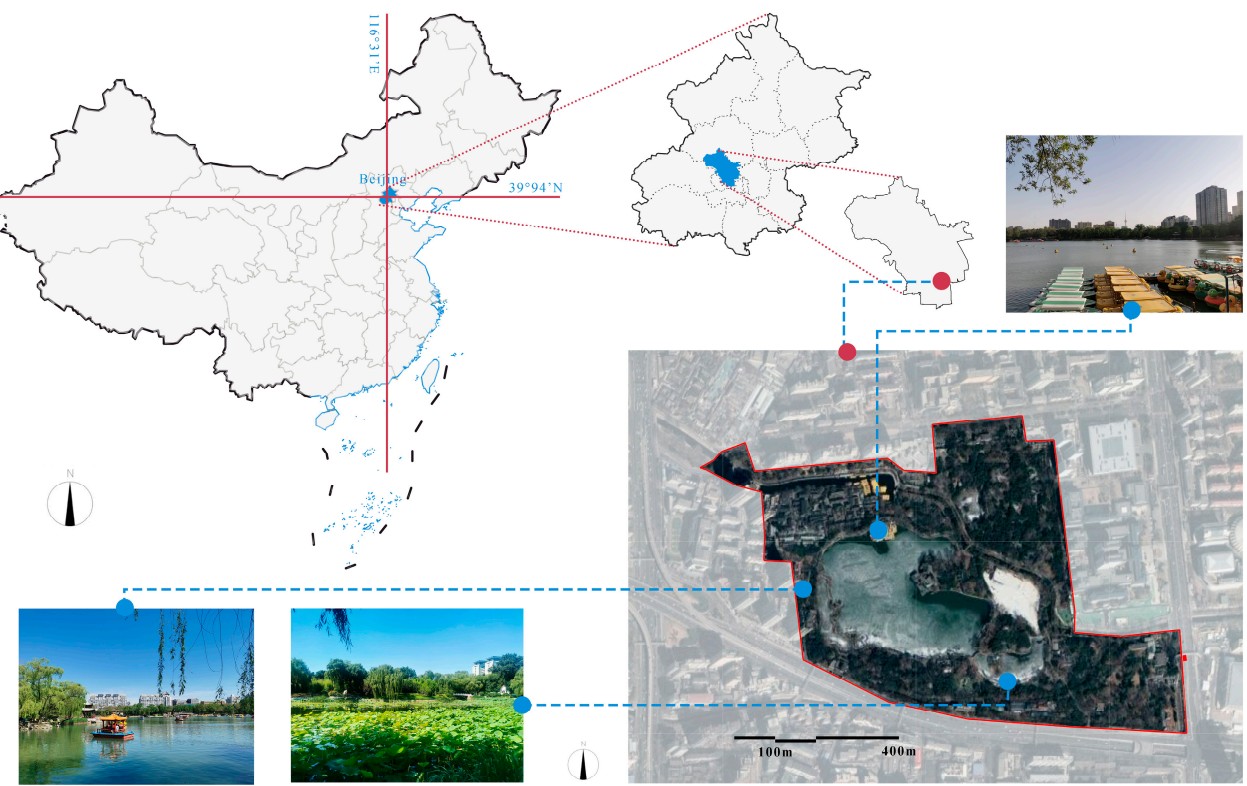

**Figure 1.** Site location.

### 2.2. *Smellwalk and Smellmap*

Sensory walking is one of the most important methods of sensory research, aiming at identifying, evaluating and recording different sensory elements through human senses [7]. The methodology and process of a smellwalk was referenced from and improved upon previous studies [42,43]. Seven investigators (average age 22 ± 2 years) with a background in landscape architecture were selected to participate in the smellwalk and had to have normal smelling abilities and to not have come in contact with food or perfume within 2 h prior to the study. The weather conditions during the investigation were stable, with a light breeze, no rain, and a temperature range of 28–35 °C, which were representative of the season. The smellwalk was conducted in the afternoon in July, and the route was a circular tour in the park, passing through various landscapes, such as wooded grasslands, waterfront trestles, and ancient buildings.

Before the smellwalk, the leader explained the research process and the use of the software to ensure that everyone could record the odors effectively. The smellwalk was led by the leader along a specific route. During the investigation, when an odor was identified, the investigator recorded the name of the odor and evaluated the intensity, dispersion range, and preference of the odor on a smellnote. Moreover, odor source locations were recorded via an app. To prevent a decrease in the olfactory sensitivity of the sniffing agents, the entire smellwalk was completed within 1 h. Odors were named and recorded on the smellnote (Figure 2) based on the source of the odor, e.g., flowers, garbage, sweat, etc. The location of the odor source was marked via the Two Steps Outdoor Assistant App (V7.1.6-1212, Shenzhen, China), which can record the route of an outdoor journey and mark certain points during the study, as a sniffingmap. The intensity, dispersion range, and preference of the odor were recorded based on individual perceptions using a seven-level scale. Moreover, other comments or thoughts related to olfactory perception, such as color, texture, etc., could also be recorded on the smellnote to better understand the odor. A detailed smellnote table was designed and referenced from a previous study [32,33].

## Smellnote

Name： ______ Date： ______

| Number | Name of smell | Smell Intensity weak——strong | Dispersion range little——wide | Smell like/ Dislike dislike—like | Comments & thoughts （color，texture） |
|--------|---------------|------------------------------|-------------------------------|----------------------------------|--------------------------------------|
| 1 | Grass | | | | Soft, comfort, green |
| 2 | ...... | | | | ...... |
| 3 | | | | | |
| 4 | | | | | |
| 5 | | | | | |
| 6 | | | | | |
| 7 | | | | | |
| 8 | | | | | |
| 9 | | | | | |

**Figure 2.** Smellnote used in smellwalks. The first row of the table is an example of odor recording. The red circle represents the score that investigators choose.

The smellmap was plotted by referring to the contour line method [32,44]. The legend of the smellmap is shown in Figure 3. Sniffingmaps of different investigators were overlaid via Photoshop (CC 2018, USA) and plotted as a smellmap. Similar smells recorded by different investigators were grouped into one type using the same color legend (Figure 3).

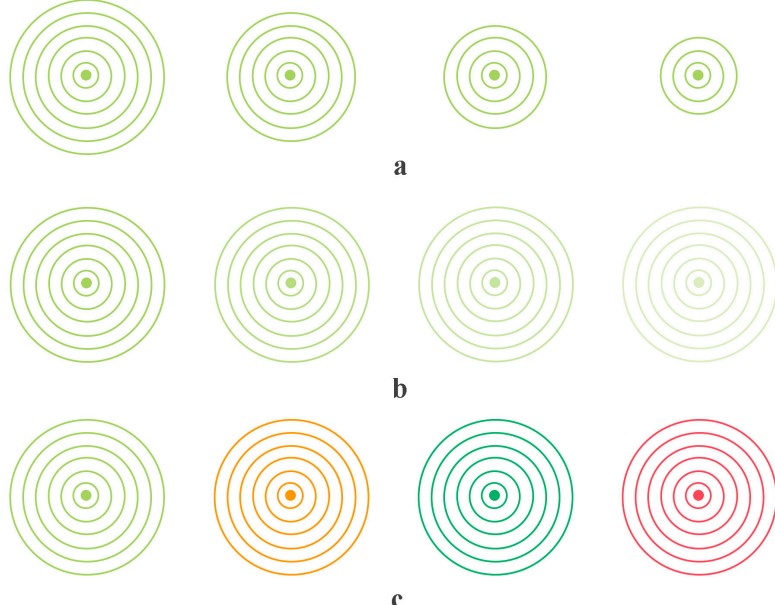

**Figure 3.** Legend of the smellmap: (**a**) the spread of the smell; (**b**) the intensity of the smell; (**c**) different types of smell. The number of circles represents the spread of the smell, the transparency represents the intensity of the smell, and the different colors represent different types of smell.

### 2.3. Smell Classification

The identification and classification of odors is always the first step in odor research. However, due to the complexity of odors, it has always been difficult to reach a consensus on how to classify them. In urban areas, odor types tend to be richer and to cover multiple categories due to the diverse human activities in a city [32,45]. Many odor classification systems have been proposed, including an urban smell classification system and an urban

park smell classification system, only typical odor types were studied in our study based on the previous field research [40]. The smells recognized in Purple Bamboo Park were classified into natural and artificial smells. Considering the historical park characteristics of the study site, the four elements of traditional Chinese garden landscapes—mountains, water, plants, and architecture—were adopted to classify odors into seven subcategories, including plants, soil, water, architecture, paving, body odor, and others (Figure 4) [46]. Since natural odors predominate in the park, plant odor classification was further divided into flowers, grass, pine, bamboo, and lotus. A total of 22 odors were identified in Purple Bamboo Park (Table 1).

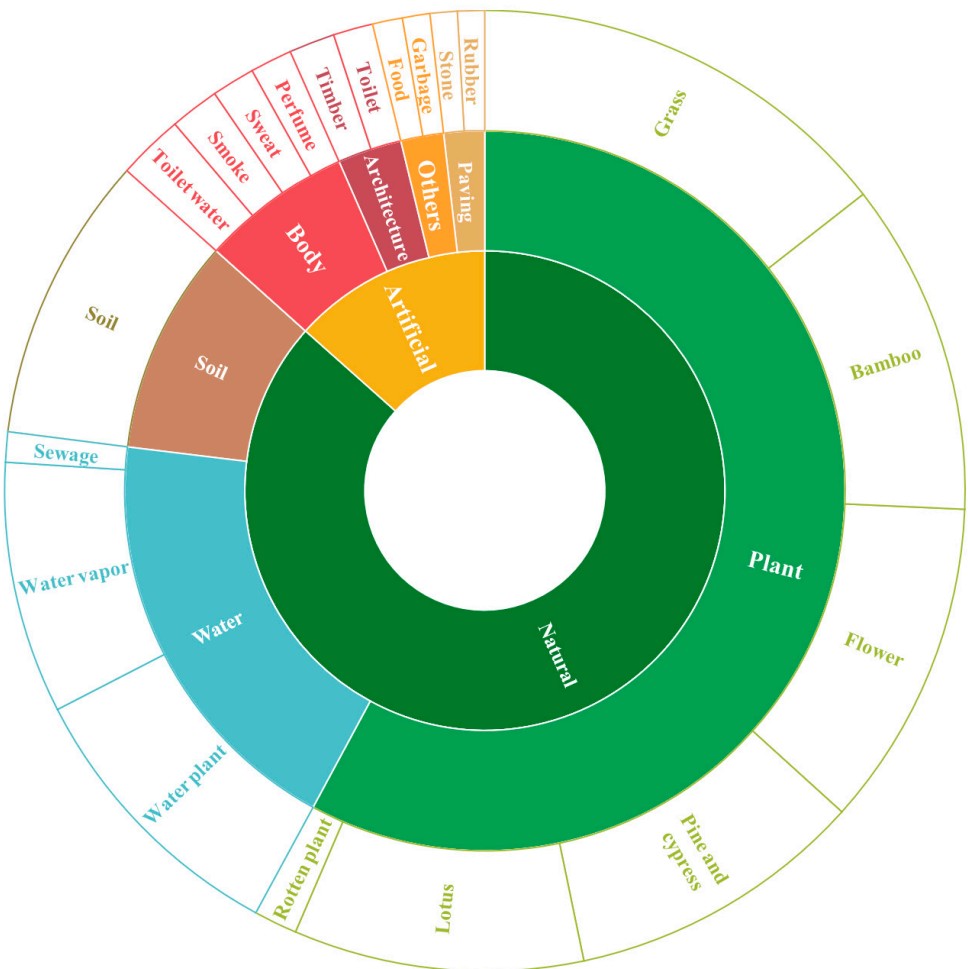

**Figure 4.** Categories of smells in Purple Bamboo Park. Top-level categories are in the inner circle; second-level categories are in the outer ring; and smell source names are in the outermost ring.

**Table 1.** Composition of typical smell sources in Purple Bamboo Park.

| Class | Subclass | Smell Source Name | |
|---|---|---|---|
| Natural | Plant | Flowers, grass, pine and cypress, bamboo, lotus | Rotten plant |
| | Soil | Soil | |
| | Water | Water plants, water vapor | Sewage |
| Artificial | Architecture | Timber | Toilet |
| | Paving | Rubber, stone | |
| | Body | Perfume, toilet water | Smoke, sweat |
| | Others | Food | Garbage, exhaust |

### 2.4. Construction of the Questionnaire

The questionnaire was used to investigate the characteristics of public olfactory perception and its relationship with visit experience. The questionnaire survey was conducted under clear and windless weather conditions throughout the park area. Visitors were randomly selected as interviewees covering different gender and age groups. Before the questionnaire, we confirmed that the participants' sense of smell was normal and affirmed that the results would not be used for purposes other than the study. Participants were not misled or disturbed in the process of completing the questionnaire, and the process used for the questionnaire survey adhered to the Code of Ethics. The questionnaire consists of the following three parts.

#### 2.4.1. Information about the Interviewees

The first part of the questionnaire was designed to collect the interviewee's social/demographical/behavioral information. Social and demographic indicators included gender (male and female), age (<18, 18–30, 31–40, 41–50, 51–60, and >60), educational background (below junior high school, junior high school, high school/secondary school, university, and graduate student and above), and occupation (with landscape architecture background and without landscape architecture background). Behavioral indicators included visit duration (<30 min, 30 min–1 h, 1 h–2 h, and >2 h) and visit frequency (first time, hardly (1–2 times a year), sometimes (once every few months), often (1–2 times a month), usually (1–2 times a week), and frequently (3 times a week and more)).

#### 2.4.2. Visiting Experience

Visiting experience consisted of three dimensions. Visual perception was the subjective evaluation of the visual aesthetics of the urban park; olfactory perception was the subjective evaluation of the olfactory comfort of the urban park; and overall perception was the subjective evaluation of overall satisfaction with the tourist experience. Interviewees were asked to evaluate the visual perception, olfactory perception. and overall perception of the urban park on a five-point Likert scale (1—very bad; 2—bad; 3—neither good nor bad; 4—good; and 5—very good).

#### 2.4.3. Individual Smell Identification

The third part of the questionnaire was about the olfactory perception of the park. Based on the preliminary investigations, 22 different smells regularly appearing in parks were selected for evaluation, including natural smells (flowers, grass, pine and cypress, bamboo, lotus, soil, water vapor, water plants, and water vapor) and artificial smells (timber, rubber, stone, perfume, toilet water, food, toilets, smoke, sweat, garbage, and exhaust).

To better depict odor characters, some metrics from the perception of sound were referenced and modified. Based on the concepts of Sound Dominant Degree and Sound Harmonious Degree, Smell Dominant Degree, Smell Harmonious Degree, Perceived Occurrence of Individual Smells, Perceived Intensity of Individual Smells, and Preference for Individual Smells were proposed [19,20].

Perceived Occurrence of Individual Smells (POS) refers to the frequency of perceiving certain odors during the entire tour in the park and was evaluated on a four-level scale (0—could not smell; 1—occasionally smelled; 2—often smelled; and 3—frequently smelled). Perceived Intensity of Individual Smells (PIS) refers to the average odor concentration perceived by the respondents and was evaluated on a five-level scale (1—very weak; 2—weak; 3—neither strong nor weak; 4—strong; and 5—very strong). Preference for Individual Smells (PFS) refers to the degree of subjective preference of respondents for smells and was evaluated on a five-level scale (1—dislike; 2—relatively dislike; 3—neither prefer nor dislike; 4—relatively prefer; 5—prefer).

Smell Dominant Degree (SDD) refers to the dominant position of certain smells, determined by their perceived occurrences and perceived intensity, as shown in Function (1).

$$SDD = POS*PLS \tag{1}$$

Smell Harmonious Degree (SHD) refers to the harmonious status of certain smells, determined by their perceived occurrences and preference, as shown in Function (2).

$$SHD = POS*PFS \tag{2}$$

### 2.5. Statistical Analysis

The questionnaire data were processed in SPSS 16.0. The primary statistical methods included Spearman's rho correlation analysis. Smellwalk data were exported via the Two Steps Outdoor Assistant App as a sniffingmap. The labeled points on the sniffingmap corresponded to the odor information recorded on the smellnote to define the odor type, concentration, and diffusion range. According to the legend (Figure 3), the odor identification points were marked with different colors, sizes, and transparency in Photoshop CC 2018. The final smellmap was plotted by overlaying the smellwalk results from all the investigators.

## 3. Results

### 3.1. Smell Distribution Characteristics

The odor background of Purple Bamboo Park was investigated through smellwalks. According to the different characteristics of each odor, smellmaps were classified into a plant smellmap, a background smellmap, and an unfavorable smellmap (Figure 5).

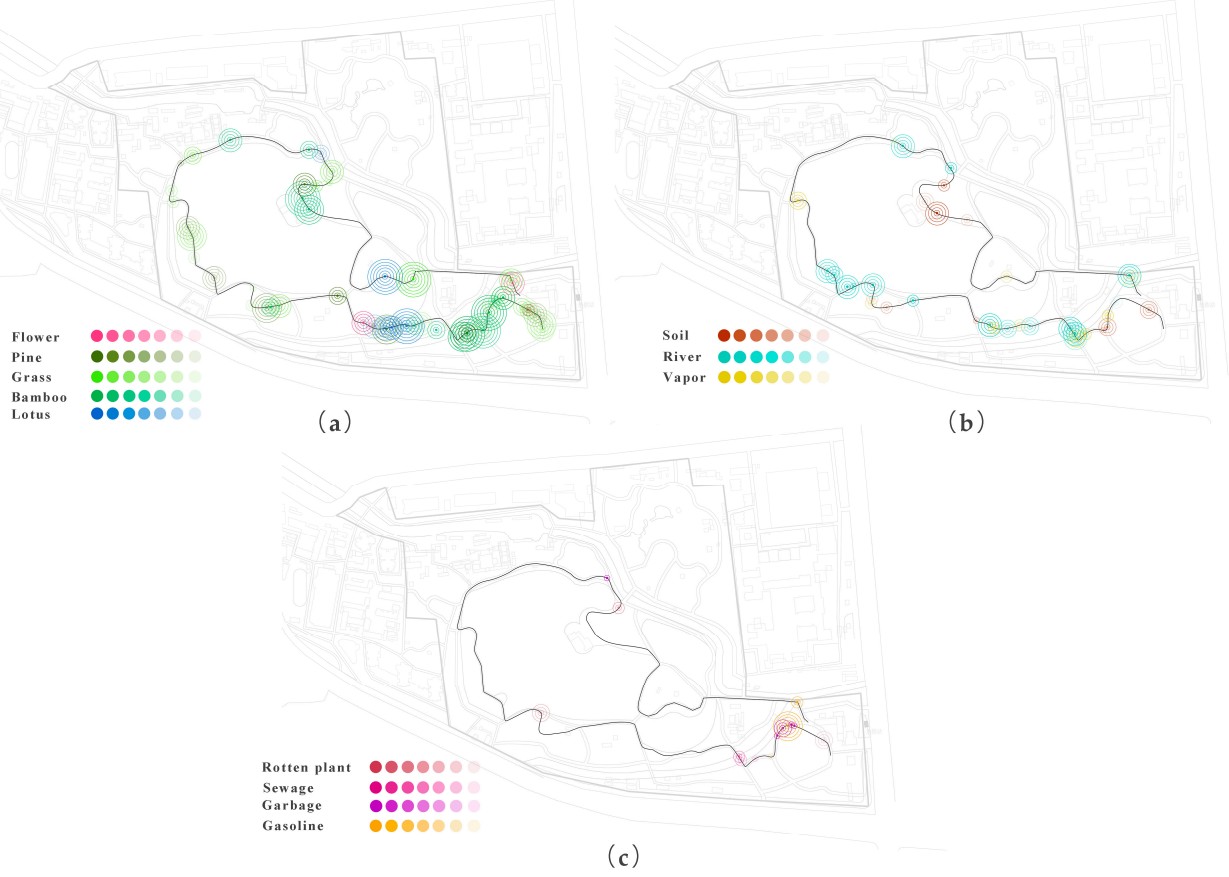

**Figure 5.** Smellmap in Purple Bamboo Park: (**a**) The plant smellmap; (**b**) the background smellmap; (**c**) the unfavorable smellmap.

Five types of plant smells were recorded in the plant smellmap, including flowers, grass, pine and cypress, bamboo, and lotus. Plant odors are evenly distributed in the park and correlate with the distribution of plant communities. As there are fewer flowering plants in northern China during the summer months, the scent of flowers occurs less frequently, while the scent of grass and bamboo occurs more frequently. Background odors included soil, river, and vapor, which set the background tone of the park. Background odors are usually low in concentration but have an extensive diffusion range. Unfavorable odors including rotten plants, sewage, garbage, and gasoline, by contrast, are usually less diffuse but more concentrated. Unfavorable odors are closely associated with human activities such as exhaust emissions and waste disposal and occurs less frequently than plant smells and background smells. Overall, odors in Purple Bamboo Park are dominated by natural odors, with background odors as a base. There are few negative odors in the park, indicating good park stewardship.

### 3.2. Individual Smell Perception Analysis

The mean values of the individual smell perception indicators for the smell sources in Purple Bamboo Park are shown in Figure 6. The statistics for interviewees' information in the park are shown in Figure 7. All natural smell sources, except rotten plants, showed higher POS values than artificial smell sources. Grass (1.86) was the most frequently perceived plant smell, followed by flowers (1.55), while lotus (1.25) showed the lowest value among plant smell. It is assumed that the planting of lotus is concentrated in specific water areas and, therefore, is perceived less frequently during the overall tour. Toilet water (0.42) showed the highest POS value among artificial smells. Rubber (0.08) and exhaust (0.08) showed the lowest value among all smells.

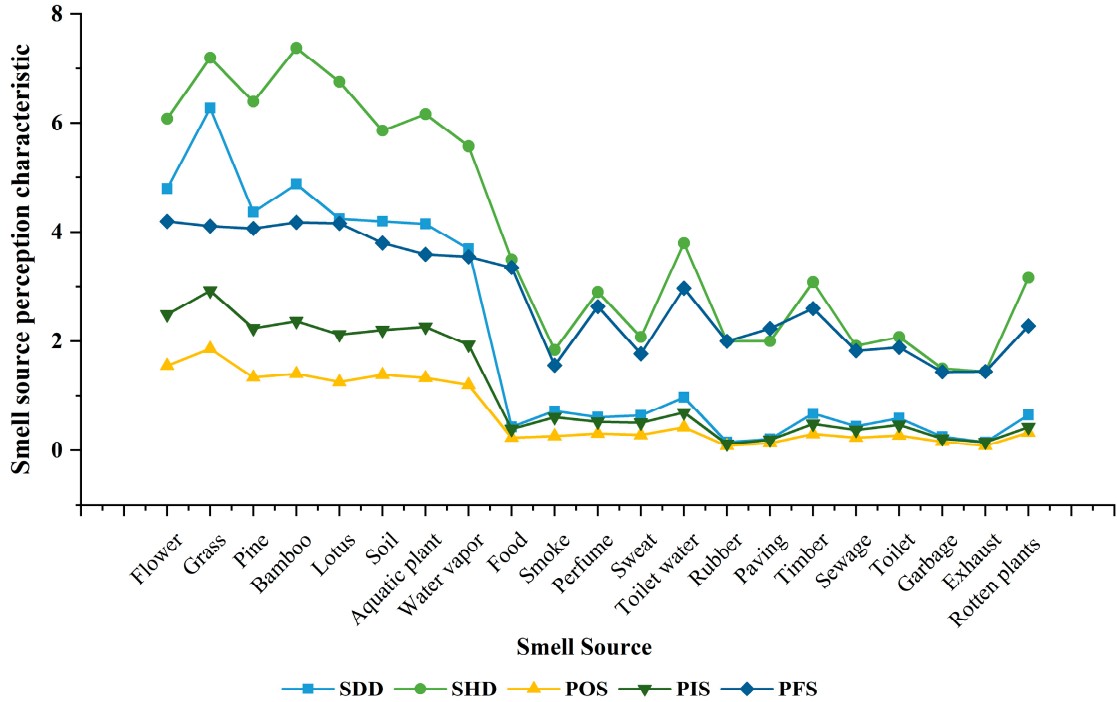

**Figure 6.** Smell source perception characteristic analysis. POS: Perceived Occurrence of Individual Smells; PIS: Perceived Intensity of Individual Smells; PFS: Preference for Individual Smells; SDD: Smell Dominant Degree; SHD: Smell Harmonious Degree.

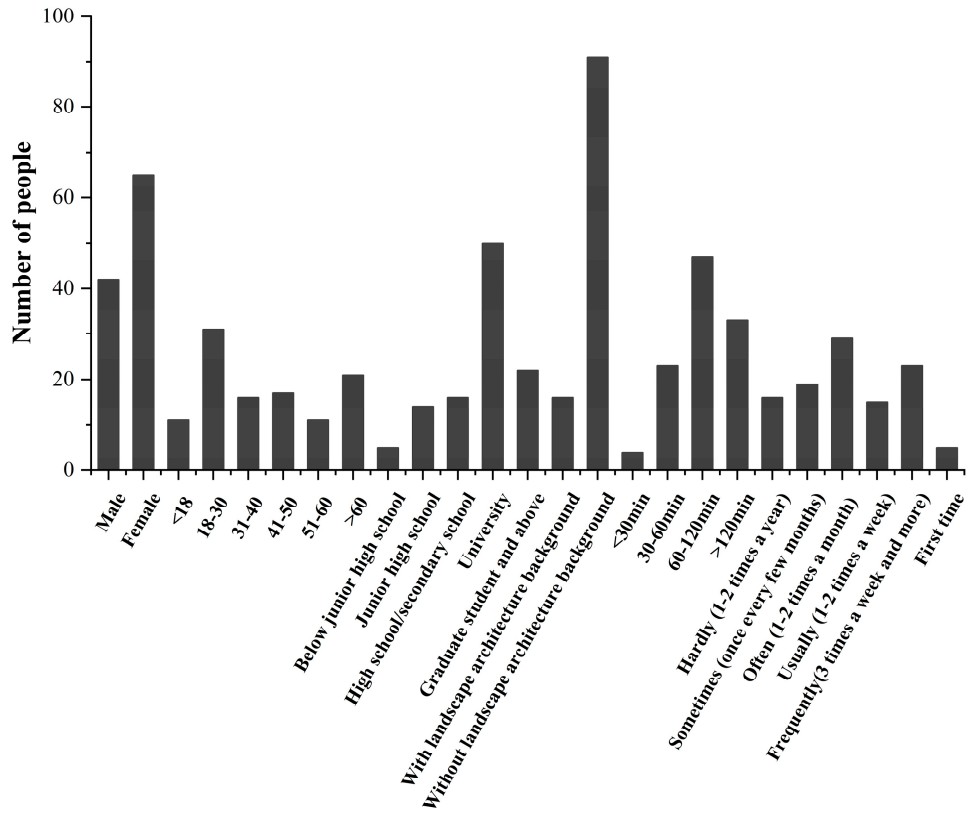

**Figure 7.** Statistics for interviewees' information in the park.

As for PIS, the first six kinds of smells receiving the highest values were all natural, showing a similar pattern to POS. Grass (2.93) showed the highest PIS value, while lotus (2.12) was the lowest one among all plant smells. Water vapor (1.93) has the lowest intensity of all natural smells, and rubber (0.12) showed the lowest value among all smells.

The PFS values for natural smells were all higher than 3, except for rotten plants (2.28), with flowers (4.19), bamboo (4.18), lotus (4.16), and grass (4.11) being the most dominant smells. Food (3.35) showed the highest PFS value, followed by toilet water (2.97), among artificial smells. Garbage (1.44), exhaust (1.44), and smoke (1.56) had the lowest preferences, and they were all typically negative smells.

In terms of SDD, grass (6.27), bamboo (4.88), and flowers (4.79) were the most dominant smell sources, while lotus (4.24) showed the lowest dominant degree among the plant smells. In addition to the four plant smells, the highest degree of dominance was for soil (4.20). It is worth mentioning that the SDD of almost all natural smells, except the smell of rotten plants, were higher than the SDD of artificial smells. Toilet water (0.97) showed the highest dominant degree among artificial smells, while exhaust (0.15) and rubber (0.15) showed the lowest dominant degree.

As for SHD, bamboo (7.37), grass (7.19), and lotus (6.75) showed the highest harmonious degree among all smells. Flowers (6.08) showed the lowest harmonious degree among plant smells. All the artificial smells showed a low harmonious degree, among which exhaust (1.44) and garbage (1.50) showed the lowest harmonious degree.

Figure 8 shows the distribution of the interviewees' Perceived Occurrence of Different Smells. With the exception of rotting plant odors, natural odors were recognized at over 60%, while artificial odors were recognized at less than 40%. Grass (91.6%) and flowers (86.9%) were the most frequently perceived smells in the green spaces, which indicates that natural smells are the keynotes of most urban park areas. It is worth mentioning that the frequency of identification of natural smells is not concentrated at one frequency but is more evenly distributed, suggesting that there are individual differences in the identification of smells. Among all the artificial smells, the smell of toilet water (31.8%)

was the most frequently identified, while the smell of rubber (7.5%) was the loudest smell frequently perceived. In general, natural smells play a more significant role in the olfactory environment of urban parks.

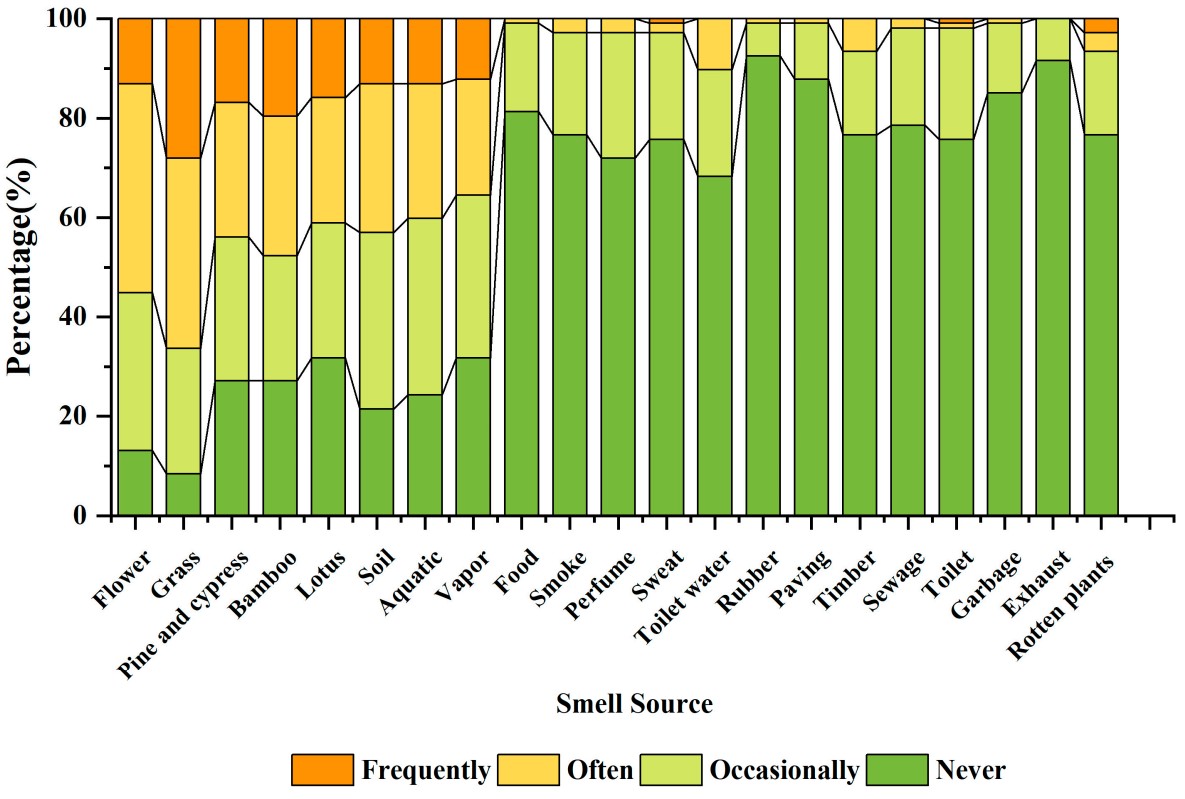

**Figure 8.** Recognition frequency of different smell sources.

### 3.3. Effects of Individual Smell Perception on Visiting Experience

3.3.1. Relationship between Smell Perception and Visual Experience Degree

The correlation between smell source perception indicators and visual experience rating based on Spearman's rho correlation analysis is presented in Table 2. A total of six smells were significantly associated with the visual experience score, all of which were plant smells. All three olfactory indicators of grass—POS, PIS, and SDD—showed significant correlations with visual experience scores, indicating that the smell of grass plays a positive role in visual experience. As for SHD, only aquatic grass showed a significant positive correlation with visual experience scores. As for PFS, all five plant smells showed a significant positive correlation with visual experience scores, with bamboo showing the highest correlation (0.338) and lotus showing the lowest correlation (0.240). Only plant smells showed a significant positive correlation with visual experience scores, with grass having the strongest relationship.

3.3.2. Relationship between Smell Perception and Olfactory Rating

Five smells showed significant correlations with olfactory experience scores: four plant smells and the smell of soil (Table 2). All five indicators of flowers showed a highly significant positive correlation with olfactory experience degree, indicating that aromatic plants play an essential role in the olfactory experience. Consequently, the layout of aromatic plants can be considered in park landscape design to improve tour experiences. Moreover, the harmony of natural smells such as lotus, pine, and cypress all showed a significant positive correlation with the olfactory rating. In addition to plant odors, soil odors are natural odor sources that present a positive correlation to the olfactory experience. The PFS of soil was significantly correlated with olfactory experience degree. In addition,

PFS and SDD both showed significant relationships with olfactory experience degree among all individual smell perception indicators, while POS and SHD showed fewer relationships than PFS and SDD.

**Table 2.** Correlation between visual experience degree, olfactory experience degree, and overall experience degree with each of the smell source perception indicators.

| Experience Degree | POS | PIS | PFS | SDD | SHD |
|---|---|---|---|---|---|
| visual experience degree | grass (0.230 *) [1] | grass (0.274 **) | bamboo (0.338 **) pine and cypress (0.322 **) aquatic (0.287 **) flowers (0.285 **) lotus (0.240 *) | grass (0.247 *) | aquatic (0.240 *) |
| olfactory experience degree | flowers (0.386 **) | flowers (0.266 **) grass (0.211 *) | lotus (0.303 **) pine and cypress (0.301 **) soil (0.227 *) flowers (0.213 *) | flowers (0.346 **) | flowers (0.368 **) lotus (0.243 *) pine and cypress (0.232 *) |
| overall experience degree | flowers (0.202 *) | grass (0.193 *) sweat (−0.195 *) | flowers (0.437 **) bamboo (0.340 **) lotus (0.304 **) pine and cypress (0.236 *) | sweat (−0.194 *) | flowers (0.361 **) |

[1] Smells showing significant relationships with experience degree are listed with the correlation coefficients in parentheses, * ($p < 0.05$) and ** ($p < 0.01$). POS: Perceived Occurrence of Individual Smells, PIS: Perceived Intensity of Individual Smells, PFS: Preference for Individual Smells, SDD: Smell Dominant Degree, SHD: Smell Harmonious Degree.

### 3.3.3. Relationship between Smell Perception and Overall Experience Degree

Six smells showed significant correlations with overall experience scores: five plant smells and one body smell (Table 2). The PFS of four plant smells showed significant relationships with overall experience degree. Similar to olfactory satisfaction, flowers remain strongly associated with overall satisfaction since the POS, PFS, and SHD of flowers all showed highly significant positive correlations with olfactory experience degree. Sweat was the only smell that showed a negative effect on the overall experience degree. It is recommended that the flow of visitors should be controlled appropriately during hot weather in summer to ensure positive visitor experiences.

To sum up, compared with artificial smells, natural smells are more relevant to the visual, olfactory, and overall experience of visits. Plant smells, such as the smell of flowers, grass, pine, cypress, and bamboo, all play a significant positive role in the visit experience. Among them, the smell of flowers, of which nine indicators are related to recreational experience, has the most significant impact on the tour experience. Moreover, PFS showed the most significant relationships with experience degree among all individual smell perception indicators. It is speculated that people who have a higher preference for natural smells are more satisfied with the urban park environment.

### 3.3.4. Correlation between Visit Experience Degrees

The correlation between visual experience degree, olfactory experience degree, and overall experience degree is shown in Table 3. The correlations between overall experience degree, visual experience degree, and olfactory experience degree are all highly significant, indicating that visual, olfactory, and overall feeling are an organic unit interconnected with each other. Although the influence of visual feeling on touring experience is stronger than that of olfactory feeling on touring experience, the sense of smell still has a great influence on touring visiting experience.

**Table 3.** Correlation between visual experience degree, olfactory experience degree, and overall experience degree.

| Experience Degree | Visual Experience Degree | Olfactory Experience Degree | Overall Experience Degree |
|---|---|---|---|
| visual experience degree | 1 | - | - |
| olfactory experience degree | 0.572 ** | 1 | - |
| overall experience degree | 0.734 ** | 0.667 ** | 1 |

** Significant at the 0.01 level (two-tailed test).

*3.4. Effects of Social/Demographical/Behavioral Factors on Individual Smell Perception*

The relationships between each of the social/demographical/behavioral factors and smell perception indicators were analyzed based on Spearman's rho correlation analysis (see Table 4). Educational background only showed a significant relationship with toilet odor. The POS (0.259), PIS (0.257), and SDD (0.257) of toilet odor all showed a significant relationship with educational background. Visit duration showed a highly significant correlation with the POS (0.198), PIS (0.235), and SDD (0.223) of soil odor. It is inferred that soil, as a background smell, was more frequently perceived as the duration of the tour increased. For both elements, educational background and visit duration, there was a link to the perception of only one odor. As for visit frequency, the POS of grass (0.205), smoke (0.195), and toilet water (0.233) showed a significant correlation with the factor. The PIS (0.254) and SDD (0.251) of toilet water showed a significant relationship with visit frequency. Moreover, the PFS of smoke (0.474) and the SHD of grass (0.200) also showed a significant relationship with visit frequency. Five perceptual indicators of three odors were associated with the factor, which indicates that visit frequency is the perceptual feature most strongly associated with social and behavioral characteristics.

**Table 4.** Correlation between each of the smell source perception indicators and social/demographic/behavioral indicators.

| Social/ Demographic/ Behavioral Indicators | POS | PIS | PFS | SDD | SHD |
|---|---|---|---|---|---|
| education | toilet (0.259 **) [1] | toilet (0.257 **) | - | toilet (0.257 **) | - |
| visit duration | soil (0.198 *) | soil (0.235 *) | - | soil (0.223 *) | - |
| visit frequency | grass (0.205 *)<br><br>smoke (0.195 *)<br>toilet water (0.233 *) | toilet water (0.254 **) | smoke (0.474 *) | toilet water (0.251 **) | grass (0.200 *) |

[1] Smells showing significant relationships are listed with the correlation coefficients in parentheses, * ($p < 0.05$) and ** ($p < 0.01$). POS: Perceived Occurrence of Individual Smells, PIS: Perceived Intensity of Individual Smells, PFS: Preference for Individual Smells, SDD: Smell Dominant Degree, SHD: Smell Harmonious Degree.

Age showed no significant relationship with any smell source perception indicators. Objectively, there should be differences in olfactory sensitivity and preferences between age groups, but in the present study, such differences did not reach statistical significance.

**4. Discussion**

*4.1. Differences in Smellscapes in Urban Parks and Urban Environments*

Odors in the city mainly include traffic emissions, industrial odors, food odors, etc., which are closely related to human production activities, while the proportion of natural odors is small. Compared with the study of urban smellscape in Beijing, there were more abundant natural odor types in the urban park. It was similarly confirmed by the results of the scent survey in London, England, where Hyde Park was the most perceptible location

for natural odors [45]. In this study, most natural odors have a recognition rate of over 60% and positively impact the visit experience. It is found that natural odors showed significance in the olfactory environment of urban parks. Natural smells enrich the visit experience and are an essential entry point for constructing park smellscapes.

In addition to this, parks have a large spread of natural smells as a smellscape background. The different natural odors are able to form an organic whole, rather than existing in fragments, interspersed with the industrial odors of the city. Olfaction is also an essential component of the multi-sensory experience. A complete smellscape can be better integrated with other sensory elements to provide a better visitor experience, such as the abundance of scented plants creating a richer experience in a multi-sensory garden. Previous studies have found that aromatic plants have a debilitating effect on the perception of noise [47]. Comforting odors can ease the discomfort of hot summer weather [48]. Our study also found highly significant correlations between visual, olfactory, and overall satisfaction, suggesting that the sense of smell interacts with the other senses. Moreover, a comfortable olfactory experience can enhance overall visitor satisfaction and contribute to the sustainability of park construction.

*4.2. Suggestions for Management and Design of Urban Park Smellscapes*

The sense of smell is an indispensable factor in the urban sensory experience. Olfactory experience can strengthen people's sense of participation and recognition at a location and can promote positive experiences with the other senses. Therefore, the establishment of a smellscape is of great significance. The design and management of a smellscape should be based on a good understanding of the smellscape in the existing environment [38]. Therefore, a background odor environment survey of the existing park environment is essential. Purple Bamboo Park was built a long time ago and has a rich landscape composition. Summer is a season for growing plants, with high temperatures and abundant odor types, making a smellscape during this period representative. Therefore, we conducted a smellwalk and questionnaire survey in the park in July to explore the existing odor environment, and a preliminary characterization of the park's smellscape was derived.

In the background of globalization, the construction of landscapes has gradually converged and lacked regional landscape characteristics. The construction of olfactory cultural landscapes is a research direction to provide tourists with a more unique tour experience, such as the Spice Market, a special smellscape in Istanbul, Turkey, increasing awareness of the importance of scents for local residents and visitors [26]. In China, the olfactory culture also has a long history. In many classical gardens, there are attractions named after the smellscape, for example, "He Feng Si Mian" refers to the smell of lotus in summer and "Xue Xiang Yun Wei" refers to the smell of plum blossom in winter [28]. This provides a reference for the construction of modern urban parks. An in-depth understanding of the smellscape characteristics of the local culture can provide a more long-term and sustainable development potential for landscape construction.

The design and management of a smellscape are not limited to controlling negative odors [49,50]. On the contrary, the management and creation of positive odors are equally important, such as the creation of night gardens and multi-sensory gardens [11,51–53]. Fragrant night-blooming plants like jasmine, tuberose, and night-blooming cereus can create a magical and enchanting atmosphere. Multi-sensory gardens are pivotal for offering immersive experiences that engage sight, touch, sound, and especially smell. They provide therapeutic benefits, promote relaxation, and foster emotional well-being. An exemplary case is the "Healing Garden" at the Cleveland Clinic, designed to soothe patients and visitors through carefully curated plants and scents [54]. Moreover, research in Alnarp Rehabilitation Garden, Sweden, suggests that the natural odors dominantly reported evoked positive emotions, memories, and joy as well as stress reductions [51]. Combined with this study, it can be found that plants are crucial for the construction of smellscapes in urban parks. Thoughtful plant selection can evoke pleasant emotions, enhancing the space's comfort and appeal. Additionally, plant scents can trigger memories, promote

relaxation, and support mental well-being. Therefore, when designing a smellscape, careful consideration of plant types and placement is essential.

There are several suggestions for smellscape management and design in urban parks: Positive smells should be protected and emphasized to maintain a park's smellscape character. The creation of smellscapes can be integrated with local cultural contexts, such as the culture of scented plants. Plant odor is an essential element in the construction of garden smellscapes. On the one hand, plant odors can be more easily controlled than artificial odors, as they can be adjusted through plant community allocation and maintenance. On the other hand, the VOCs of plants can contribute to a healing experience. Furthermore, park services need to clean up trash and thin out foot traffic to ensure a clean environment in their parks. In addition, hosting a scent sniffing experience program could be considered to give people new perspectives on experiencing the park, which could bring people together, fostering a sense of community and shared experience.

## 5. Conclusions

The olfactory experience is an essential part of the multi-sensory experience of urban parks. In this study, we conducted a smellwalk and questionnaire survey in Purple Bamboo Park in Beijing to investigate the types, classifications, distributions, and publicly perceived characteristics of odors in urban parks. The results showed that natural odors dominated the urban park smellscape. Moreover, all olfactory perception characteristics significantly correlated with landscape satisfaction degree. Among all these factors, PFS was the most associated with landscape satisfaction. Flowers and grass are the odors that have the most decisive influence on tour experience. The results of this study reveal the influence of olfactory perception on visiting experience and offer a new perspective on the multi-sensory experience in gardens. It provides a theoretical basis and suggestions for managing and constructing smellscape in urban parks. Further, it also provides some reference for the design of sensory experiences in urban public spaces. However, the research results can only represent the current situation regarding olfactory perception in parks in the summer, and more cases are needed to further improve the research on olfactory perception in the future. Further research thus is needed to compare smellscapes for different types of parks or to explore the changing patterns of smellscapes across seasons. It would also be an interesting direction to explore the differences in the perception of smellscapes across different cultural contexts.

**Author Contributions:** Conceptualization, C.W. and J.Z.; methodology, C.W.; software, C.L., H.L. and B.T.; validation, C.W. and R.Z.; formal analysis, C.W. and H.S.; investigation, C.W., H.S., R.X., L.X. and X.Y.; resources, C.W.; data curation, R.Z.; writing—original draft preparation, C.W. and R.Z.; writing—review and editing, C.W. and J.Z.; visualization, C.W.; supervision, M.S.; project administration, M.S.; funding acquisition, M.S. All authors have read and agreed to the published version of the manuscript.

**Funding:** This research was funded by the National Natural Science Foundation of China (31971708, 32271947); the Beijing Natural Science Foundation (6202022); and the Science, Technology & Innovation Project of Xiongan New Area (2022XAGG0100).

**Institutional Review Board Statement:** Not applicable.

**Informed Consent Statement:** Not applicable.

**Data Availability Statement:** Data are contained within the article.

**Acknowledgments:** We thank all investigators from the summer practice group "Seasonal smellscape" in Beijing Forestry University for their contribution to data collection.

**Conflicts of Interest:** The authors declare no conflict of interest.

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
