# Peer review of "Smellscape Characteristics of an Urban Park in Summer: A Case Study in Beijing, China"

_sustainability, doi:10.3390/su16010163_

Round 1

Reviewer 1 Report

Comments and Suggestions for Authors

Dear authors,

Here are my comments for improvement:

Introduction:

1. Consider expanding on the broader context of the significance of multisensory experiences in public spaces, emphasizing how smellscape contributes to the overall perception of these environments.

2. Including recent studies or trends related to the importance of olfactory experiences in urban settings could further strengthen the introduction and highlight the relevance of the study in the current research landscape.

Materials and Methods:

1. It would be important to elaborate on the specific criteria for participant selection and the process of conducting the smellwalk and questionnaire survey to enhance the transparency and replicability of the study.

2. Adding information about the tools and techniques used for data analysis would further strengthen the methods section and provide readers with a better understanding of the analytical process.

Results:

1. Provide a more detailed discussion of the statistical analyses and any significant patterns observed among different demographic groups could offer a deeper understanding of the relationships between smell perception and various factors.

Discussion:

1. The discussion effectively highlights the importance of smellscape design in urban parks and the potential impact of natural odors on visitor experiences. It would be useful to integrate more examples or case studies from other urban parks or gardens to provide a broader perspective on the management and construction of smellscape in various settings.

2. Also, consider expanding on the implications of the findings for park design and management, including practical recommendations for enhancing the olfactory environment, would strengthen the discussion and provide actionable insights for park planners and designers.

Conclusions:

1. Provide more explicit recommendations for future research directions, such as exploring olfactory perceptions in different seasons or conducting comparative studies in other urban park contexts.

2. Highlighting the broader implications of the study for the field of environmental psychology or urban planning could provide a more holistic perspective on the significance of smellscape research in the broader context of public space design and management.

Good luck!

Comments on the Quality of English Language

English fine

Author Response

Response to comments by Reviewer #1

Introduction:

  1. Consider expanding on the broader context of the significance of multisensory experiences in public spaces, emphasizing how smellscape contributes to the overall perception of these environments.

Re: Thanks for your suggestions. We have added the significance of multisensory experiences in public spaces, and emphasized how smellscape contributes to the overall perception of these environments in the introduction part.

  1. Including recent studies or trends related to the importance of olfactory experiences in urban settings could further strengthen the introduction and highlight the relevance of the study in the current research landscape.

Re: Several recent studies have been added in the revised manuscript, and the importance of olfactory experiences in urban settings was further emphasized.

Materials and Methods:

  1. It would be important to elaborate on the specific criteria for participant selection and the process of conducting the smellwalk and questionnaire survey to enhance the transparency and replicability of the study.

Re: More details of the specific criteria for participant selection and the process of conducting the smellwalk and questionnaire survey were added in the Methods part.

  1. Adding information about the tools and techniques used for data analysis would further strengthen the methods section and provide readers with a better understanding of the analytical process.

Re: We have added more detailed introductions of tools and techniques used for data analysis, especially for the odor mapping.

Results:

  1. Provide a more detailed discussion of the statistical analyses and any significant patterns observed among different demographic groups could offer a deeper understanding of the relationships between smell perception and various factors.

Re: We have added more discussions to highlight the key findings and offer a deeper understanding of the relationships between smell perception and various factors.

Discussion:

  1. The discussion effectively highlights the importance of smellscape design in urban parks and the potential impact of natural odors on visitor experiences. It would be useful to integrate more examples or case studies from other urban parks or gardens to provide a broader perspective on the management and construction of smellscape in various settings.

Re: Thanks for your kindly suggestions. We have integrate more examples or case studies from other urban parks or gardens in the second part of discussion,and hope to provide a broader perspective on the management and construction of smellscape in various settings.

  1. Also, consider expanding on the implications of the findings for park design and management, including practical recommendations for enhancing the olfactory environment, would strengthen the discussion and provide actionable insights for park planners and designers.

Re: Based on the suggestions, we have discussed the management and design of smellscapes in more detail and provide actionable insights for park planners and designers in the second part of discussion.

Conclusions:

  1. Provide more explicit recommendations for future research directions, such as exploring olfactory perceptions in different seasons or conducting comparative studies in other urban park contexts.

Re: As suggested by the reviewer, we have added future perspectives of the research to the conclusion section. Including how smellscapes differ in different types of parks and seasons.

  1. Highlighting the broader implications of the study for the field of environmental psychology or urban planning could provide a more holistic perspective on the significance of smellscape research in the broader context of public space design and management.

Re: Thanks for your kindly suggestions. The implications of this study have been strengthened accordingly in the revised manuscript.

Besides suggestions from Reviewers, we tried our best to improve the manuscript and made some changes in the manuscript. These changes will not influence the content and framework of the paper. And here we did not list the changes but marked in red in revised manuscript.

Thank you very much for your comments and suggestions.

Reviewer 2 Report

Comments and Suggestions for Authors

The paper approaches integrated urban development from a fascinating and non-conventional point of view. The concept of smellscape is introduced through the multi-sensory landscape development. The case study was made in a public park in Beijing.

The methodology is clearly described. Although smell sense is quite subjective, the use of complex methods is capable of making the research more objective. The results are comprehensively delineated. Unfortunately, the discussion chapter is relatively short, it should be broadened. The conclusions part is too short, it does not contain the future perspectives of the research. As smellscapes are culturally constructed phenomena, it would be interesting to examine how people, with different cultural backgrounds can evaluate the same location.

Author Response

Response to comments by Reviewer #2

The paper approaches integrated urban development from a fascinating and non-conventional point of view. The concept of smellscape is introduced through the multi-sensory landscape development. The case study was made in a public park in Beijing.

The methodology is clearly described. Although smell sense is quite subjective, the use of complex methods is capable of making the research more objective. The results are comprehensively delineated.

1. Unfortunately, the discussion chapter is relatively short, it should be broadened.

Re: Thanks for your suggestions. We have strengthened the discussion section to address the differences between urban odors and park odors as well as the recommendations for managing and designing of park smellscapes.

2. The conclusions part is too short, it does not contain the future perspectives of the research.

Re: Future perspectives of this study have been expanded appropriately, including how smellscapes differ in different types of parks and seasons.

3. As smellscapes are culturally constructed phenomena, it would be interesting to examine how people, with different cultural backgrounds can evaluate the same location.

Re: Thank you for your suggestions, but unfortunately cultural backgrounds were not addressed in this study. However, we have added this point to the future perspectives of the research and hope to explore it in the future study.

Besides suggestions from Reviewers, we tried our best to improve the manuscript and made some changes in the manuscript. These changes will not influence the content and framework of the paper. And here we did not list the changes but marked in red in revised manuscript.

Thank you very much for your comments and suggestions. 

Reviewer 3 Report

Comments and Suggestions for Authors

This is an interesting work and generally well detailed and well written. I read with intrigue and especially enjoyed the robust presentation of results. However, there is one major area for improvement and two smaller ones. The first is a major flaw of the work and requires attention.

The "smellscape" concept is intriguing and supported. However, there needs to be a social science theory or conceptual framework underpinning this work. This was notably missing from the background, as were research questions that then specifically related the smellscape to this theory/framework for application in the authors' study site. See previous work on other sensory attributes, such as "soundscapes," for examples on how there is a sensory component as the main focus but there are still groundings in particular theories/frameworks as justification for the study and its contribution beyond the context. These would then be points to follow on in the Discussion, for placement of the work in the larger body of knowledge.

It would be good to know how the smells composition and extent differed on the smellwalk versus other areas of the park or the nearby surroundings. That would give a sense of the degree of attractiveness of such a trail and how much the smellscape here was different from the baseline conditions of someone's experience in Beijing. (Note: I thought this is where the first paragraph of the Discussion was going, but it was instead only park focused.)

It appears that the "Institutional Review Board" statement is missing. This is concerning, as there were survey respondents. If ethical permissions are not required for studies in this context, please clarify.

Comments on the Quality of English Language

Generally fine. Minor issues throughout that can be attended to in a routine final copy edit.

Author Response

Response to comments by Reviewer #3

This is an interesting work and generally well detailed and well written. I read with intrigue and especially enjoyed the robust presentation of results. However, there is one major area for improvement and two smaller ones. The first is a major flaw of the work and requires attention.

1. The "smellscape" concept is intriguing and supported. However, there needs to be a social science theory or conceptual framework underpinning this work. This was notably missing from the background, as were research questions that then specifically related the smellscape to this theory/framework for application in the authors' study site. See previous work on other sensory attributes, such as "soundscapes," for examples on how there is a sensory component as the main focus but there are still groundings in particular theories/frameworks as justification for the study and its contribution beyond the context. These would then be points to follow on in the Discussion, for placement of the work in the larger body of knowledge.

Re: Thanks for your kindly suggestions. We've added some background information about sensory component as the main focus in both introduction part and discussion part, and several essential references have been added to support the theory of "smellscape".

2. It would be good to know how the smells composition and extent differed on the smellwalk versus other areas of the park or the nearby surroundings. That would give a sense of the degree of attractiveness of such a trail and how much the smellscape here was different from the baseline conditions of someone's experience in Beijing. (Note: I thought this is where the first paragraph of the Discussion was going, but it was instead only park focused.)

Re: We've added a discussion about how different the smellscape here is from the baseline conditions of someone's experience in Beijing in the first part of the discussion. Given that smellwalks can only explore odors in the travel route, it is difficult to explore odors outside of the route, but we can determine the attractiveness of the trail directly by the presence of surrounding odor sources.

3. It appears that the "Institutional Review Board" statement is missing. This is concerning, as there were survey respondents. If ethical permissions are not required for studies in this context, please clarify.

Re: Thanks for your kind reminder. Ethical permissions are not required for such simple non-interventional studies, several similar research published on Sustainability also indicate that [1-3]. Although ethical permissions are not applicable for this study, ethical statement has also been added to the revised manuscript.

1. Hao, J.; Ye, X.; Yu, C.; Liu, J.; Ruan, Y.; Zhang, Y.; Hong, F.; Zhang, D. A Novel Individual Carbon Emission Evaluation and Carbon Trading Model for Low-Carbon University Campuses. Sustainability (Basel, Switzerland) 2023, 15, 15928.

2. Li, Z.; Yang, Y. Determinants of College Students’ Online Fragmented Learning Effect: An Analysis of Teaching Courses on Scientific Research Software on the Bilibili Platform. Sustainability-Basel 2023, 15, 16023.

3. Zhou, X.; Zhang, F.; Shan, L.; Lin, C. Research on the Current Situation and Problems of Ecological Civilization Education for Contemporary College Students—An Empirical Analysis Based on Structural Equation Modeling. Sustainability-Basel 2023, 15, 16051.

Besides suggestions from Reviewers, we tried our best to improve the manuscript and made some changes in the manuscript. These changes will not influence the content and framework of the paper. And here we did not list the changes but marked in red in revised manuscript.

Thank you very much for your comments and suggestions. 

Round 2

Reviewer 3 Report

Comments and Suggestions for Authors

My concerns have been addressed in the review. However, I would suggest that the authors continue to provide further connection to social science concepts and theories when conducting their work and presenting this paper. As often occurs, this paper centers the methods and findings at the expense of connection to a guiding framework. The authors have made progress in the Introduction in this regard, and I give them credit for it and thus this revision meets my approval for minor revisions.

Comments on the Quality of English Language

Generally fine - can use a final copy editing before publication

Author Response

Response to comments by Reviewer #3

My concerns have been addressed in the review. However, I would suggest that the authors continue to provide further connection to social science concepts and theories when conducting their work and presenting this paper. As often occurs, this paper centers the methods and findings at the expense of connection to a guiding framework. The authors have made progress in the Introduction in this regard, and I give them credit for it and thus this revision meets my approval for minor revisions. 

Re: Thank you for your valuable comments. They are very helpful for revising and improving our paper. We have studied comments carefully and have made corrections which we hope to meet with approval. Specifically, further connections to social science concepts and theories have been added in the introduction, method, and discussion parts.